# Structural insights into drug transport by an aquaglyceroporin

Wanbiao Chen[1,10], Rongfeng Zou[2,10], Yi Mei[3,4,5,10], Jiawei Li[1,6], Yumi Xuan[1], Bing Cui[1,7], Junjie Zou[2], Juncheng Wang[8], Shaoquan Lin[9], Zhe Zhang ®[3,4,5] ✉ & Chongyuan Wang ®[1] ✉

Pentamidine and melarsoprol are primary drugs used to treat the lethal human sleeping sickness caused by the parasite *Trypanosoma brucei*. Cross-resistance to these two drugs has recently been linked to aquaglyceroporin 2 of the trypanosome (TbAQP2). TbAQP2 is the first member of the aquaporin family described as capable of drug transport; however, the underlying mechanism remains unclear. Here, we present cryo-electron microscopy structures of TbAQP2 bound to pentamidine or melarsoprol. Our structural studies, together with the molecular dynamic simulations, reveal the mechanisms shaping substrate specificity and drug permeation. Multiple amino acids in TbAQP2, near the extracellular entrance and inside the pore, create an expanded conducting tunnel, sterically and energetically allowing the permeation of pentamidine and melarsoprol. Our study elucidates the mechanism of drug transport by TbAQP2, providing valuable insights to inform the design of drugs against trypanosomiasis.

Trypanosomes are protozoan parasites that cause human sleeping sickness and animal trypanosomiasis[1,2]. Pentamidine and melarsoprol have been used as anti-trypanosomatid drugs for over 70 years and remain crucial therapeutic options[3]. Pentamidine and other diamidine drugs typically accumulate to very high concentrations in the trypanosome's mitochondrion, where they bind to the kinetoplast DNA, inhibiting both replication and transcription[4–6]. However, the mechanism of action of melarsoprol and other arsenical drugs remains unclear[7]. Cross-resistance between melarsoprol and pentamidine (melarsoprol-pentamidine cross-resistance, MPXR) was observed following the introduction of these drugs, leading to 20–30% treatment failure[8]. Subsequent studies have suggested that this cross-resistance results from lower uptake rates of these drugs rather than from

mutations in their targets[9]. The first gene implicated in MPXR was the aminopurine transporter TbAT1/P2[10]. Subsequently, two additional transporters were described as high-affinity pentamidine transporter (HAPT1) and low-affinity pentamidine transporter (LAPT1)[11,12]. HAPT1 was identified as TbAQP2 through a genome-scale RNA interference screen and was found to be the primary determinant of MPXR (Fig. 1a)[10,13–16].

Aquaporin family, comprising aquaporins and aquaglyceroporins, are major intrinsic proteins that facilitate the passive transport of water, glycerol, and other small solutes[17]. *T. brucei* encodes three aquaglyceroporins, namely AQP1, AQP2, and AQP3, exhibiting a broad permeability profile for small solutes, including water, glycerol, methylglyoxal, L-lactate, D-lactate, and acetate[18]. Sequence analysis

[1]Center for Human Tissues and Organs Degeneration, Faculty of Pharmaceutical Sciences, Shenzhen Institute of Advanced Technology, Chinese Academy of Sciences, Shenzhen 581055, China. [2]Shenzhen Jingtai Technology Co., Ltd. (XtalPi), Shenzhen 518000, China. [3]Jiangsu Province Key Laboratory of Anesthesiology, Xuzhou Medical University, Xuzhou, Jiangsu Province 221004, China. [4]Jiangsu Province Key Laboratory of Anesthesia and Analgesia Application Technology, Xuzhou Medical University, Xuzhou, Jiangsu Province 221004, China. [5]NMPA Key Laboratory for Research and Evaluation of Narcotic and Psychotropic Drugs, Xuzhou Medical University, Xuzhou, Jiangsu Province 221004, China. [6]Department of Geriatric Medicine, Shenzhen Longhua District Central Hospital, Shenzhen 518110, China. [7]School of Basic Medicine and Clinical Pharmacy, and State Key Laboratory of Natural Medicines, China Pharmaceutical University, Nanjing 210009, China. [8]Advanced Medical Research Institute, Shandong University, Jinan, China. [9]Centre for Polymers in Medicine, Faculty of Pharmaceutical Sciences, Shenzhen Institute of Advanced Technology, Chinese Academy of Science, Shenzhen 581055, China. [10]These authors contributed equally: Wanbiao Chen, Rongfeng Zou, Yi Mei. ✉e-mail: zhangzhe70@xzhmu.edu.cn; cy.wang@siat.ac.cn

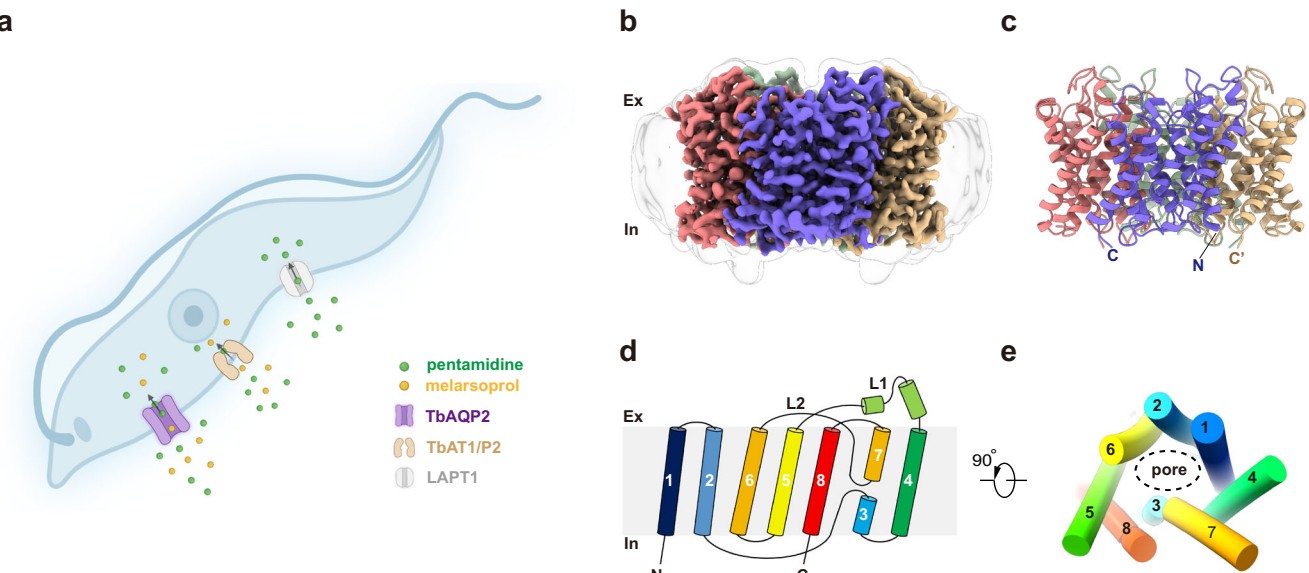

**Fig. 1 | Structure of TbAQP2. a** Schematic representation of membrane transporters involved in the uptake of trypanocides pentamidine and melarsoprol. There are three membrane transporters involved in pentamidine uptake. The high-affinity pentamidine transporter (HAPT1, identified as TbAQP2), the P2 aminopurine transporter (TbAT1/P2), and the low-affinity pentamidine transporter (LAPT1), with Km values of ~36 nM, ~430 nM and ~56 μM, respectively. TbAQP2/HAPT1 is the main contributor to pentamidine sensitivity in Trypansome. Panel A created with BioRender.com and released under a Creative Commons Attribution-NonCommercial-NoDerivs 4.0 International license. **b, c** 3D cryo-EM reconstruction (**b**) and structure (**c**) of TbAQP2 in an apo-state viewed parallel to the membrane, with subunits colored separately. **d** Topology of TbAQP2. **e** cartoon representation of the TbAQP2 monomer colored the same as in d, viewed extracellularly. In **e**, helices are shown as cylinders.

revealed that TbAQP1 and TbAQP3 encode canonical selectivity filters, whereas TbAQP2 possesses a unique selectivity filter[15]. Previous studies have indicated that TbAQP2 transports pentamidine and melarsoprol via direct permeation[16,19,20]. However, melarsoprol and pentamidine exhibit significant structural differences compared to typical AQP substrates. While water (18 Da), urea (60 Da), and glycerol (92 Da) are typical AQP substrates with hydrophilic properties, melarsoprol (398 Da) and pentamidine (340 Da) have much larger molecular weights and predominantly hydrophobic and dicationic organic properties, respectively. These substantial differences have sparked debates regarding the 'drug channel' hypothesis of TbAQP2. An alternative proposition, the 'porin-receptor' hypothesis, suggests that drug uptake occurs through binding to TbAQP2, followed by endocytosis[21]. To investigate the mechanism of TbAQP2-dependent uptake of anti-trypanosomatid drugs, we determine high-resolution cryo-electron microscopy (cryo-EM) structures of TbAQP2 in its substrate-free and drug-bound states. Our structural analysis unveils that pentamidine and melarsoprol are accommodated in wide conducting tunnels through hydrophobic interactions and hydrogen bonds. Comparisons with canonical water- or glycerol-permeable AQPs indicate that multiple substitutions in pore-lining residues and rearrangements near the extracellular entrance of TbAQP2 result in an expanded conducting pore, enabling the accommodation and permeation of pentamidine and melarsoprol.

## Results

### Functional reconstitution and structure determination

To investigate the transport activity of TbAQP2 in a simpler manner, we developed the cell-based uptake assay using the fluorescent analog of pentamidine (stilbamidine) as a substrate[7]. In contrast to uninfected HEK293 cells, those infected with TbAQP2 displayed robust uptake of stilbamidine (~60% positive) within 1 min (Supplementary Fig. 1a, b). Thus, TbAQP2-dependant uptake of stilbamidine in mammalian cells recapitulates the rapid uptake of pentamidine observed in Trypanosomes.

For cryo-EM studies, full-length TbAQP2 was expressed in HEK293 cells, subsequently purified, and reconstituted into lipid nanodiscs (Supplementary Fig. 1c–e). We determined high-resolution 3D reconstructions of TbAQP2 alone and in complex with pentamidine or melarsoprol, at overall resolutions ranging from 3.0 to 2.45 Å (Supplementary Figs. 2–4). These reconstructions facilitated the building of atomic models with accurate stereochemistry that correlated well with the observed cryo-EM density (Supplementary Table 1). TbAQP2 exhibited the canonical aquaporin fold, featuring eight transmembrane helices (TM1-TM8) connected by six loops (loop A-loop F), and with both the N and C termini positioned on the cytoplasmic side of the membrane (Fig. 1b–e). Four TbAQP2 monomers assembled and formed a tetramer with a fourfold symmetry, closely resembling other aquaporins (Fig. 1b–c).

### Conducting pore and selectivity filter

Each monomer of TbAQP2 features a conducting pore that extends ~25 Å from the extracellular vestibule to the intracellular vestibule. (Fig. 2a–d). The substrate selectivity of AQPs is thought to be controlled by a selectivity filter (SF) situated below the extracellular vestibule (Fig. 2a–d)[22–26]. In conventional AQPs, a distinctive feature known as the aromatic/arginine motif (ar/R) in the selectivity filter plays a crucial role in determining selectivity. For channels selective to water, the selectivity filter consists of four highly conserved residues (F58, H182, C191, and R197 in bovine AQP1) (Fig. 2a, e). The histidine residue projects towards the pore, constricting its diameter to less than 2.0 Å (Fig. 2a, d, e), effectively preventing the passage of glycerol or other solutes. In glycerol-permeable channels, also known as aquaglyceroporins, the selectivity filter is formed by three bulky residues (W50, F190, and R196 in PfAQP from the malarial parasite Plasmodium falciparum), with the side chain of phenylalanine pointing outward from the pore (Fig. 2b, d, f). This outward orientation relieves the constriction to approximately 3.5 Å, allowing the passage of glycerol (Fig. 2b, d, f). TbAQP2, however, deviates from the typical aquaglyceroporin by lacking the ar/R motif found in canonical AQPs

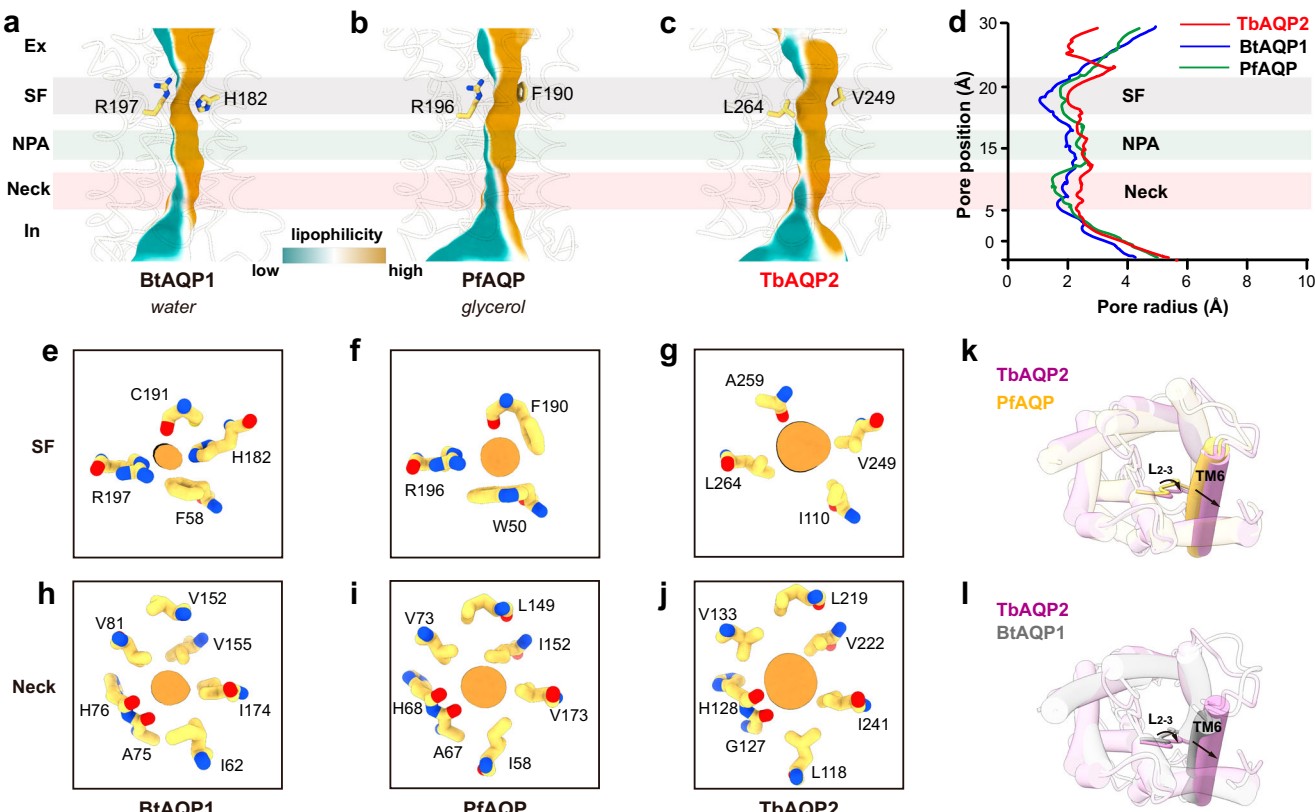

**Fig. 2 | Conducting pore. a–d** Channel profiles (**a–c**) and radii (**d**) along the pore for BtAQP1 (bovine, PDB: 1J4N), PfAQP (*Plasmodium falciparum*, PDB: 3C02) and TbAQP2, calculated using the program MOLE, selectivity filter residues are shown as sticks. The channel profiles are colored by lipophilicity generated by ChimeraX: brown means lipophilic; and green means hydrophilic. The regions for SF, NPA motifs, and neck are highlighted in gray, light green, and pink, respectively. **e–g** SF of BtAQP1 (**e**), PfAQP (**f**), and TbAQP2 (**g**), cross sections of channel profile and key residues are shown. **h–j** Necks of BtAQP1 (**e**), PfAQP (**f**), and TbAQP2 (**g**), cross sections of channel profile and key residues are shown. **k, l** Superpositions of TbAQP2 (pink) with PfAQP (yellow), TbAQP2 with BtAQP1 (gray), view from the extracellular side. Arrows indicate the shift of TM6 and loop F ($L_{2-3}$).

and the bulky residues observed in glycerol-permeable PfAQP. (Supplementary Fig. 5). Contrary to conventional AQPs, TbAQP2 features a distinct motif (I110, V249, A259, and L264) in its selectivity filter. This unique motif creates a significantly wider and more hydrophobic selectivity filter, providing steric and hydrophobic environments conducive to the passage of larger organic molecules, such as pentamidine and melarsoprol (Fig. 2c, d, g). Our structural analysis aligns with this observation. Notably, earlier studies demonstrated that mutating these selectivity filter residues to larger counterparts (I110W or L264R) completely abolished pentamidine uptake into trypanosome parasites[20].

In canonical AQPs, a distinctive "fireman's grip-like" structure is formed in the middle of the conducting pore by two Asn-Pro-Ala motifs (NPA/NPA) (Supplementary Fig. 6a, b). The conserved asparagine (Asn, N) residues within these motifs play a crucial role in orienting water during channel traversal and preventing proton transport[22–26]. TbAQP2, on the other hand, exhibits a similar fold with the motifs NS131A/NPS263 (Supplementary Fig. 6c). Interestingly, a hydrogen bond between the hydroxyl group of S263 and N261, not observed in other AQPs, is formed in TbAQP2 (Supplementary Fig. 6a–c). While these residues, S131 and S263, do not directly line the pore through which pentamidine passes (Supplementary Fig. 6c), their role in substrate binding is evident. When these unique NSA/NPS motifs in TbAQP2 are replaced with canonical NPA/NPA motifs, there is a significant ~20-fold decrease in pentamidine uptake, indicating the involvement of these serine (S) residues in substrate binding[20]. Notably, although S131 and S263 might not directly interact with pentamidine, replacing S263 with alanine disrupts the hydrogen bond to N261,

resulting in impaired substrate binding and transport (Supplementary Fig. 6c and Supplementary Movie).

In addition to differences in the selectivity filter and NPA motif, a significant distinction lies in the "neck" region between the NPA motif and the intracellular vestibule of TbAQP2 compared to other AQPs. The neck of TbAQP2 is notably wider (with a radius of ~2.5 Å) than that of water-specific or glycerol-permeable channels (with radii of ~1.5 Å) (Fig.V). Structural superposition indicates that the expansion of the neck is not caused by alterations in the pore-lining residues but rather by an outward rearrangement of TM5 and the loop connecting TM6 and TM7 (loop F) (Fig. 2k, i and Supplementary Fig. 7d). In canonical AQPs, TM5 and loop F are stabilized by a delicate hydrogen bond network involving highly conserved residues (such as E144, T189, and G190 in BtAQP1 from the bovine, Bos taurus, or E141, T188, and G189 in PfAQP from the malarial parasite Plasmodium falciparum) (Supplementary Fig. 7a, b). However, in TbAQP2, loop F becomes decoupled from TM5, leading to the outward reorganization of loop F and TM5 and, consequently, the expansion of the neck (Supplementary Fig. 7). Overall, these structural differences, including substitutions in residues lining or near the pore, contribute to an expanded selectivity filter and neck in TbAQP2, favoring the binding and subsequent permeation of pentamidine and melarsoprol.

## Pentamidine binding

To gain further insights into the mechanism of substrate binding and permeation, cryo-EM studies were conducted on TbAQP2 in the presence of drug substrates. Through optimized sample preparation procedures, structures of the channel bound to pentamidine or

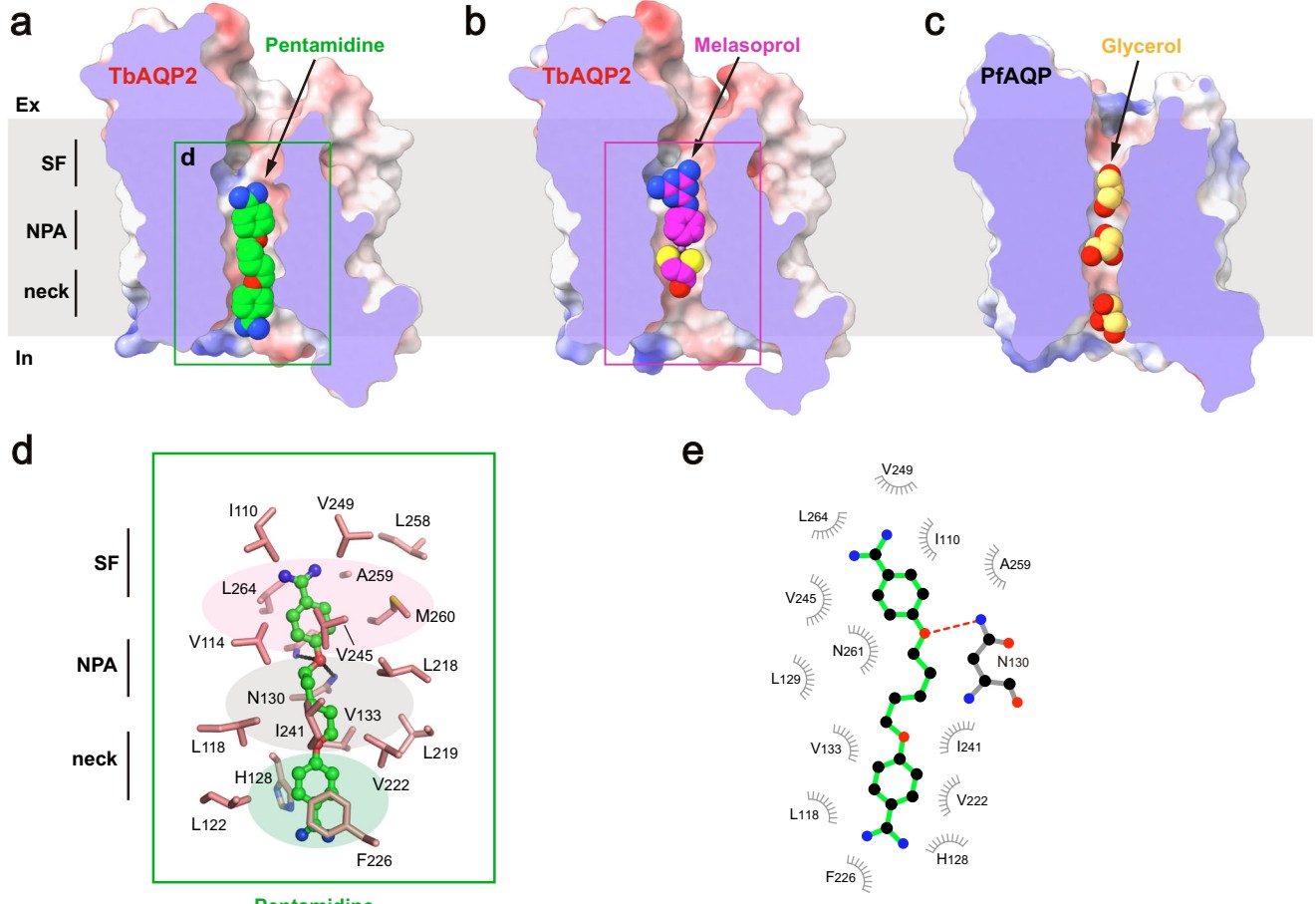

**Fig. 3 | Pentamidine binding. a−c** Cross sections of the TbAQP2 and PfAQP, showing pentamidine, melarsoprol, and glycerol (shown as spheres) bound in the conducting pores. **d** Detailed views of pentamidine bound in the pore. Pentamidine is shown in stickball models, and interacting residues are shown as sticks. **e** LigPLOT[43] diagram of TbAQP2:pentamidine interactions showing hydrogen bond interactions (dashed lines) and hydrophobic contacts ≤ 5 Å.

melarsoprol were determined at resolutions of 2.45 Å. Clear cryo-EM densities inside the conducting pore corresponding to the substrates were identified (Supplementary Fig. 3h). By comparing these structures with the apo-state structure, it was discovered that neither pentamidine nor melarsoprol induces significant conformational changes in the channel (root mean square deviation from the apo-state structure of 0.25 Å² or 0.37 Å², respectively) (Supplementary Fig. 8). Due to their relatively large sizes, only single pentamidine or melarsoprol molecules were observed inside the pore (Fig. 3a, b), in contrast to a line of water or glycerol molecules observed in the pore of canonical AQPs (Fig. 3c), suggesting a distinct permeation mechanism for these antimicrobials.

Pentamidine was found to be bound in a highly extended conformation, with one of the amidine groups entering the intracellular vestibule of TbAQP2 (Fig. 3a). This suggests that the pentamidine-bound structure represents a pre-entry state. The binding tunnel involves the selectivity filter, NPA motif, and neck. The upper benzamidine moiety of pentamidine was bound in a pocket near the selectivity filter, formed by a series of hydrophobic residues (I110, V114, M260, V245, V249, L258, A259, and L264) (Fig. 3a, d, e). The pentanediol moiety was accommodated by residues surrounding the NPA motif (L118, V133, L218, L219, V222, and I241) (Fig. 3a, d, e). In addition to hydrophobic interactions, the amide groups of N130 and A259 formed hydrogen bonds with the oxygen atom of the pentanediol moiety (Fig. 3d, e). This finding helps explain previous observations that substitutions of the ester groups with thioether or amide groups significantly decrease the uptake efficiency of these compounds[20].

Furthermore, the lower benzamidine moiety of pentamidine engaged in π-stacking interactions with F226 and H128, further contributing to high binding affinity (Fig. 3d, e). Aligning with our structural observations, mutagenesis of these substrate-binding residues resulted in significant decreases in TbAQP2-dependent uptake of stilbamidine (Supplementary Fig. 9).

It is worth mentioning that D265, previously implicated in the direct binding of pentamidine in the 'porin-receptor' mode[21], is not within direct bonding distance to pentamidine in our cryo-EM structure. Instead, it forms a salt bridge with a neighboring R269 (Supplementary Fig. 10). This suggests that the reduced pentamidine internalization in the D265A mutant is likely due to an allosteric effect that alters packing, electrostatic, and hydrogen bonding networks critical for maintaining the 'fireman's grip' of TbAQP2 and promoting substrate binding and internalization.

**Melarsoprol binding**

In contrast to the binding site of pentamidine, which was located on the intracellular side of the pore, melarsoprol was accommodated in the middle of the pore (Fig. 3b). To minimize steric clashes, melarsoprol appeared to adopt an approximately planar conformation, with the melamine moiety close to the extracellular side and the dithiasrolane moiety pointing towards the intracellular side (Figs. 3b and 4a, b). The melamine moiety of melarsoprol was bound within the selectivity filter through hydrophobic packing interactions with the pore-lining residues I110, M196, V249, and L264, and a hydrogen bond with the carbonyl group of A259 (FigV). The dithiasrolane moiety of

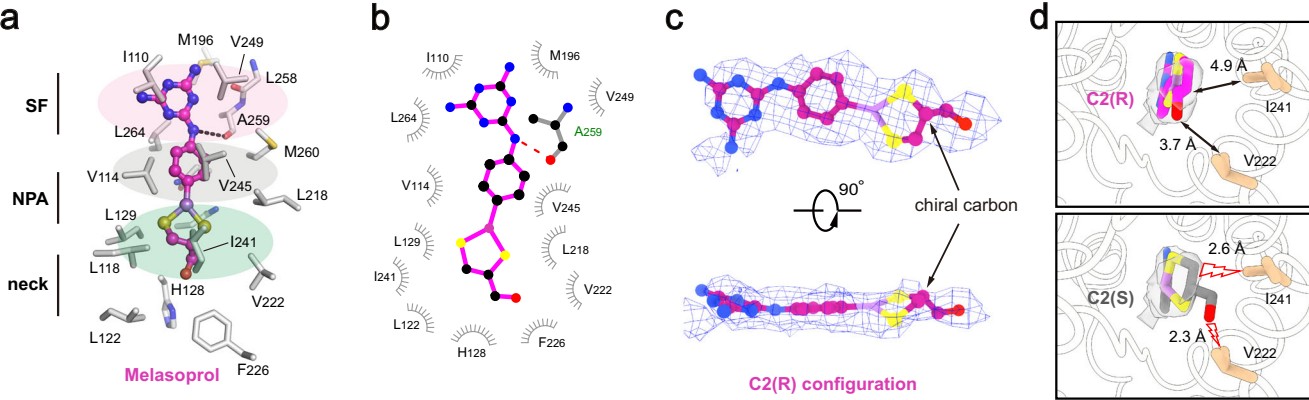

**Fig. 4 | Melarsoprol binding. a** The detailed view of TbAQP2-melarsoprol interactions. Melarsoprol is shown in ball-and-stick model, and the surrounding residues are shown as sticks. **b** LigPLOT[43] diagram of TbAQP2-melarsoprol interactions, showing the hydrogen bond interactions (dashed lines) and hydrophobic contacts (≤5 Å). **c** the bound melarsoprol adopts C2(R) configuration. Cryo-EM densities are shown as meshes, and the chiral cartoon of melarsoprol is highlighted with arrows. **d** Interactions between the carbinol group of melarsoprol and the surrounding residues of TbAQP2. Model in either C2(R) configuration (upper panel) or C2(S) configuration (lower panel) was docked into the density of melarsoprol (semi-transparent rendering). The distances between carbinol groups and surrounding residues are labeled and highlighted with arrows or lightning plots.

melarsoprol was buried in a flat hydrophobic pocket formed by residues L118, L129, V222, and I241 (Fig. 4a, b).

We identified a chiral carbon (C2 atom) in the dithiasrolane moiety of melarsoprol, presenting two possible chiral configurations (Fig. 4c, d). During cryo-EM sample preparation, a mixture of melarsoprol diastereomers was added to the TbAQP2 sample with a ~20-fold molecular excess (see Methods). The high-quality cryo-EM density inside the pore exhibited excellent correlation with one melarsoprol molecule in the C2(R) configuration (Fig. 4c). To explore the impact of chirality at the C2 atom on melarsoprol binding, we modeled a molecule in the C2(S) configuration into the same binding site (Fig. 4d, lower panel). Structural analysis revealed the emergence of severe steric clashes between the carbinol group of melarsoprol and nearby pore-lining residues (V222 and I241) (Fig. 4d, lower panel). Collectively, these results indicate configuration-selective binding of melarsoprol through TbAQP2.

## Molecular dynamic simulations

To explore the permeation pathway of pentamidine, we conducted umbrella sampling (US) simulations to derive free-energy profiles. Initially, steered molecular dynamics simulations (SMD) were employed to pull pentamidine from the binding site in two directions along the conducting pore, mimicking both association and dissociation processes. These simulations generated the necessary windows for the subsequent US simulation. Within the US simulation, a harmonic potential was applied to restrain the ligand in each window, and free-energy profiles were then reconstructed using the Weighted Histogram Analysis Method (WHAM). Our results indicated that the pentamidine binding site corresponds to the minimum free-energy location within the conducting pore (Fig. 5a), consistent with our structural observations. Specifically, our simulations revealed that the energy barrier for pentamidine to exit the channel towards the cytoplasm was approximately 10 kcal/mol, while the barrier to exit towards the extracellular side was around ~14 kcal/mol. Notably, experimental studies have indicated a strong dependence of pentamidine uptake on the membrane potential in trypanosomes (approximately −125 mV for *T. brucei*[27]. To investigate the impact of membrane potential on pentamidine permeation, we conducted simulations in the presence of a negative membrane voltage. Notably, the entry energy barrier towards the cytoplasm decreased to ~5 kcal/mol, while the exit energy barrier towards the extracellular side increased to ~24 kcal/mol (Fig. 5a). These findings suggest that, in the presence of membrane potential, pentamidine has a greater tendency to enter the cytoplasm than to exit the

cell. This aligns with experimental observations showing significant accumulation of pentamidine at mM levels inside trypanosome parasites, and no detectable pentamidine efflux when extracellular drug was removed[4,28]. Our simulations have revealed that in the presence of membrane potential, pentamidine exhibits a notable preference for entering cells over exiting, as indicated by the differences in energy barriers. This distinctive behavior shares some similarities with inwardly rectifying ion channels, such as the mitochondrial calcium uniporter[29].

Residue W192, situated near the extracellular vestibule of TbAQP2, seems to play a role in the uptake of pentamidine, as evidenced by significantly reduced pentamidine uptake upon mutagenesis[20]. Despite the absence of direct interactions between pentamidine and W192 in our cryo-EM structure, a detailed analysis of simulation results identified a local minimum free-energy site in the pentamidine permeation route (Fig. 5a, b). In this structural snapshot, pentamidine adopts an extended conformation, with one amidine group entering the selectivity filter (Fig. 5b). The other amidine group is positioned in the extracellular vestibule, forming a pi-pi interaction with W192 (Fig. 5b). To further probe the interaction between W192 and pentamidine, we utilized MM/GBSA simulation to calculate the interaction energies of pentamidine with wild-type TbAQP2 and the W192A mutant, using the local minimum captured in the US simulation as the starting coordinate. The binding energy of the W192A mutant to pentamidine decreased by 4.2 ± 0.2 kcal/mol compared to the wild-type protein. Therefore, our calculations suggest that W192 likely serves as a docking site for pentamidine during its permeation through TbAQP2.

## Discussion

Aquaporins are a class of transmembrane proteins that facilitate the passive transport of water, glycerol, and other small solutes[17]. To date, TbAQP2 is the only aquaporin reported to uptake drug-like molecules, and it is a key determinant of cross-resistance between melarsoprol and pentamidine in human trypanosomosis. However, the uptake mode of these anti-trypanosomatid drugs has been under debate, with both the 'drug channel' and 'porin-receptor' hypotheses proposed to explain the underlying mechanism[16,19–21]. Our structural studies and molecular dynamic simulations strongly support the 'drug channel' hypothesis.

Melarsoprol and pentamidine markedly differ from typical AQP substrates due to their larger size and increased hydrophobicity, suggesting a distinct mechanism underlying their permeation. Our

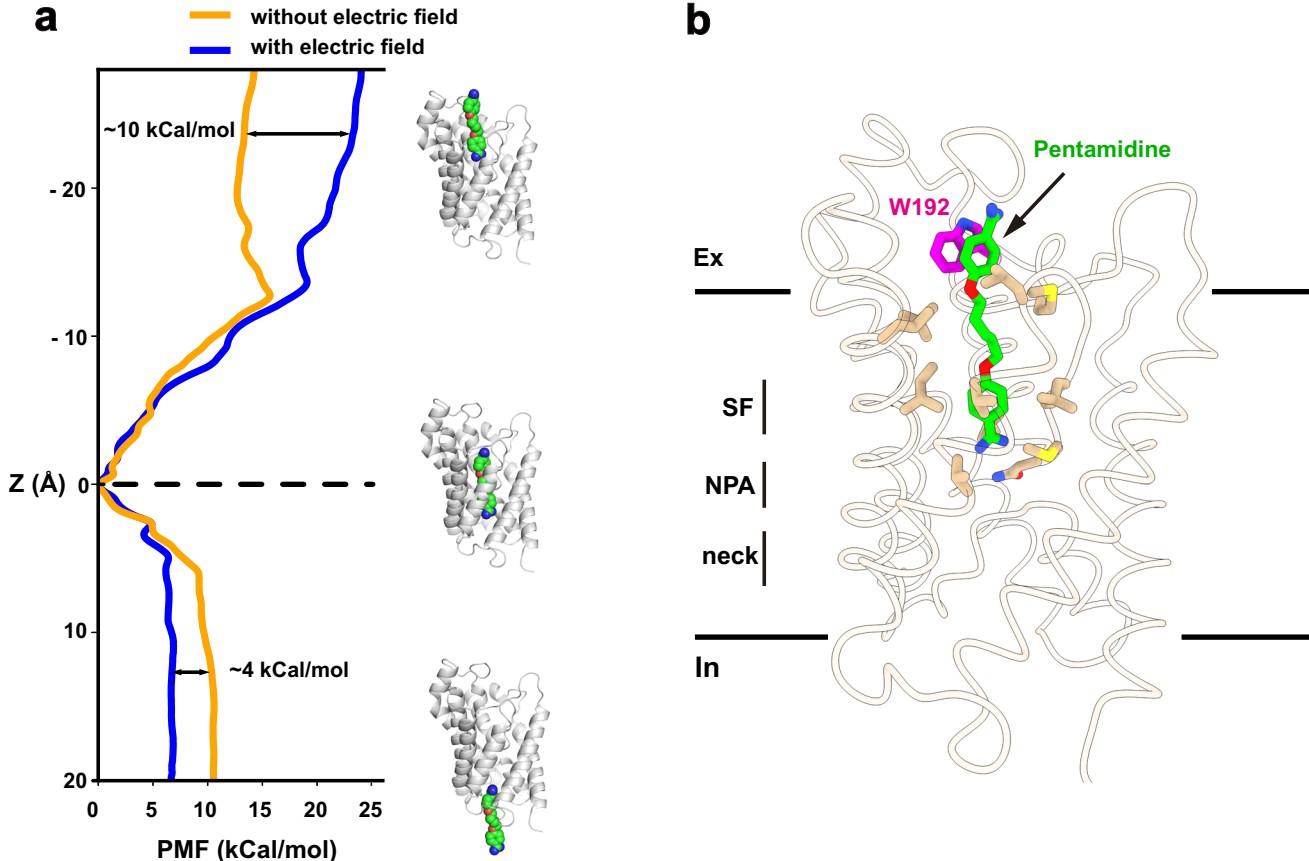

**Fig. 5 | Molecular dynamic simulations. a** Free-energy profile of pulling pentamidine to the cytoplasm and the extracellular, from the bound position, with (blue line) or without (yellow line) of an electrical field. **b** Residue W192 involving pentamidine binding and permeation. Cartoon representation of the local minima on the protein-ligand binding path. Pentamidine (green) and W192 (magentas) were shown in sticks.

structural analyses reveal variations at multiple residues lining or near the pore, resulting in a wide and hydrophobic conducting pore. This configuration allows for the sterically and energetically favorable permeation of pentamidine and melarsoprol. Furthermore, our resolved structures for TbAQP2-drug complexes, in which melarsoprol and pentamidine were captured in extended conformations within the conducting pore, likely representing the pre-entry state, elucidate the mechanism of substrate binding, permeation, and mutation-associated resistance.

Pentamidine uptake in *T. brucei* is markedly hindered by the knockdown of genes encoding plasma membrane H[+]-ATPases responsible for maintaining the plasma membrane potential or by the addition of ionophores (including carbonyl cyanide m-chlorophenyl hydrazone [CCCP], nigericin, and gramicidin). This observation indicates that a negative membrane potential inside the cell may act as the driving force for pentamidine uptake[3,14,20].

In line with this hypothesis, our simulations clearly demonstrate that the energy barrier for entering the cytosol dramatically decreases (by $4.2 \pm 0.2$ kcal/mol) when a negative membrane voltage is applied. In contrast to pentamidine uptake, melarsoprol uptake appears to be insensitive to membrane potential[20], suggesting that high binding affinity and a concentration gradient are the main driving forces. Despite melarsoprol having the highest efficiency among all melaminophenyl arsenicals used clinically in treatment against human trypanosomiasis, post-treatment reactive encephalopathy occurs in 5–10% of all patients treated with melarsoprol, leading to an overall fatality rate of ~50% for the patients occurred encephalopathy[3,30]. Thus, there is an urgent need to design melarsoprol analogs with milder side effects, using our structure of the TbAQP2-melarsoprol complex as a

starting point. It will be worth investigating if the introduction of positively charged groups into melarsoprol creates membrane potential-motivated force and boosts its uptake by TbAQP2.

Our work elucidates the molecular mechanism underlying the aquaporin-dependent uptake of the trypanocides pentamidine and melarsoprol. This mechanism is distinct from the receptor-mediated uptake observed for the trypanocide suramin[31]. The findings presented here also offer a framework for the potential design of analogs or trypanocides. This could pave the way for improved treatment options against trypanosomiasis.

## Methods

### Protein expression and purification

*Trypanosoma brucei* Aquaglyceroporin 2 (UniProt: Q6ZXT3) was evaluated using the fluorescence-detection size-exclusion chromatography (FSEC) screening technique[32]. cDNA encoding full-length TbAQP2 was synthesized (Sangon Biotech (Shanghai) Co., Ltd.), and ligated into a mammalian cell expression vector[32] to encode a protein containing a C-terminal twin-strep tag. The expression plasmid was transfected into HEK293S GnTI- cells (ATCC, # CRL-3022) for transient expression. In brief, 1 mg of plasmid and 3 mg of PEI40K (YEASEN, #40816ES.) were mixed in 100 mL Freestyle 293 media (Gibico, #12338018), incubated at room temperature for 30 min, and the mixture was added to 1 L of HEK293S GnTI- cells (~2.0 × 10[6] cells/mL) in Freestyle 293 media supplemented with 1% FBS (TransSerum, #PS301-02). After incubation at 37 °C for 16 h, 10 mM sodium butyrate (Coolaber, #CS9931-100g) was added, and the cells were cultured at 30 °C for another 48 h before harvesting. For purification, the cell pellet from 1 L of culture was resuspended in 40 mL buffer A (40 mM HEPES pH

7.4, 150 mM NaCl, 0.15 mg/mL DNase I, 1.5 µg/mL Leupeptin, 1.5 µg/mL Pepstatin A, 1 mM AEBSF, 1 mM Benzamidine, and 1 mM PMSF). Cells were disrupted by homogenization on ice. The membrane pellet was harvested by centrifugation at $70,000 \times g$ at 4 °C for 30 min, then resuspended in 40 mL buffer A. The membrane was solubilized by adding n-dodecyl-β-d-maltopyranoside (DDM, Anatrace) to a final concentration of 1%, and stirred at 4 °C for 1 h. Solubilized proteins were separated from the insoluble fraction by centrifugation at $70,000 \times g$ at 4 °C for 40 min, and the supernatant was filtered through a 0.45-µm polystyrene membrane (NEST). Two milliliters of STarm Streptactin Beads 4FF (Smart-Lifesciences, #SA092100) were incubated with the sample with agitation at 4 °C for 1 h. The beads were washed with 50 mL buffer B (20 mM HEPES pH 7.4, 150 mM NaCl, 1 mM DDM). TbAQP2 protein was eluted by incubating buffer B containing 5 mM biotin (Beyotime), and further purified by size-exclusion chromatography (SEC) on a Superose 6 increased, 10/300 GL column (GE Healthcare) equilibrated with buffer B. Mutations were generated using Q5 Site-Directed Mutagenesis Kit (New England BioLabs Inc.) and verified by DNA sequencing. The mutant proteins were expressed and purified using a procedure similar to that of the wild-type protein.

## Nanodisc reconstitution

The fractions obtained from size-exclusion chromatography (SEC) were combined and concentrated to ~0.5 mL (~1 mg/mL) using a Vivaspin 2 concentrator (100 kDa cutoff). This concentrated sample was then mixed with nanodisc scaffold protein (MSP1D1, 5 mg/mL, in a buffer containing 20 mM Tris-HCl, pH 7.8, 100 mM NaCl, 0.5 mM EDTA, and 5 mM sodium cholate) along with a lipid/DDM mixture (17 mM DDM, 10 mM lipids: POPE (Avanti): POPC (Avanti) with a 1:1 weight ratio) at a molecular ratio of 4:5:200 (TbAQP2: MSP1D1: lipid). After incubating for 1 h at 4 °C, ~250 mg of wet Bio-Beads SM2 (Bio-Rad) were added, and the sample was rotated at 4 °C overnight to remove detergent. To eliminate empty nanodiscs, the sample underwent further purification using STarm Streptactin Beads 4FF. In brief, the nanodisc sample was bound to 0.3 mL Streptactin Beads with rotation at 4 °C for 30 min, washed with 10 mL buffer C (20 mM HEPES, pH 7.4, 150 mM NaCl), and then eluted with 1 mL buffer C containing 5 mM biotin. The nanodisc samples were further purified by SEC (Superose 6 Increased, 10/300 GL column) in 20 mM HEPES, pH 7.4, 150 mM NaCl. The peak fractions were collected, concentrated to around 1 mg/mL (using Vivaspin 2, 100 kDa cutoff), and immediately used for cryo-EM grid preparation. For the preparation of the TbAQP2-drug complex, pentamidine (Sigma, from a 20 mM stock in DMSO) or melarsoprol (Toronto Research Chemicals, from a 20 mM stock in DMSO) was added to the TbAQP2 sample to a final concentration of ~200 µM. The mixture was then incubated for 30 min on ice before cryo-EM grid preparation.

## EM sample preparation and data acquisition

For cryo-EM grid preparation, 4 µL of the purified sample was applied to glow-discharged (10 s) Quantifoil R 1.2/1.3 grids (Au 400; Electron Microscopy Sciences). The grids were then plunge-frozen in liquid nitrogen-cooled liquid ethane using a Vitrobot Mark IV (Thermo Fisher Scientific). The Vitrobot was operated at 4 °C, with a blotting time of 2–4 s, using a blot force of '0', and maintaining 100% humidity. Micrographs were collected with a Titan Krios microscope (Thermo Fisher Scientific) operating at 300 kV, equipped with a K3 Summit detector (Gatan) in super-resolution mode. Details of all datasets can be found in Supplementary Table S1. All datasets underwent processing using the same general workflow, as outlined below.

## Cryo-EM structure determination

Supplementary Figs. 2 to 4 show cryo-EM workflows. Image processing was performed in cryoSPARC v.2[33] and RELION 3.1[34]. Movie stacks were gain-corrected, twofold binned, motion-corrected, and dose-weighted in MotionCor2[35]. Contrast transfer function (CTF) estimates were performed using Patch CTF estimation in cryoSPARC v.2, and micrographs with CTF fit resolutions better than 3.5 Å were selected using Manually Curate Exposures in cryoSPARC v.2. Particles were auto-picked using 2D Template picker in cryoSPARC v.2[33].

For the dataset of TbAQP2 without a substrate, 1,127,412 particles were extracted from 836 micrographs in cryoSPARC v.2 with a binning factor of 3 for ab initio reconstruction and heterogeneous refinements. The particles were subjected to one round of heterogeneous refinement in cryoSPARC v.2 to remove erroneously picked particles. Selected particles (325,497) were reextracted with a binning factor of 1 and subjected to several rounds of heterogeneous refinement (using C1 symmetry) to remove those particles that did not yield high-resolution reconstructions. After the heterogeneous refinement procedure, 153,589 particles were selected, reextracted, and subjected to non-uniform refinement with C4 symmetry, which yielded a reconstruction at 3.7 Å overall resolution. After one round of Bayesian polishing in RELION 3.1, the particles were subjected to CTF (both global and local) and non-uniform refinements in cryoSPARC v.2, resulting in a map at 3.0 Å overall resolution. The final reconstruction was further improved by density modification using the two unfiltered half-maps with a soft mask in Phenix[36], and then resampled to a pixel size of 0.756 Å.

For the dataset of the TbAQP2-pentamidine complex, 6,944,526 particles were extracted from 3,998 micrographs in cryoSPARC v.2 with a binning factor of 3 for ab initio reconstruction and heterogeneous refinement. The particles were subjected to one round of heterogeneous refinement in cryoSPARC v.2 to remove erroneously picked particles. Selected particles (1,883,669) were reextracted with a binning factor of 1 and subjected to several rounds of heterogeneous refinement (using C1 symmetry) to remove those particles that did not yield high-resolution reconstructions. After the heterogeneous refinement procedure, 990,813 particles were selected, reextracted, and subjected to 3D classification in RELION 3.1. 539,330 particles from the two best classes were selected and subjected to non-uniform refinement with C4 symmetry, which yielded a reconstruction at 2.9 Å overall resolution. After one round of Bayesian polishing in RELION 3.1, the particles were subjected to CTF (both global and local) and non-uniform refinements in cryoSPARC v.2, resulting in a map at 2.6 Å overall resolution. Another round of Bayesian polishing, and followed CTF (both global and local) and non-uniform refinement in cryoSPARC v.2 improved the resolution to 2.45 Å. The final reconstruction was further improved by density modification using the two unfiltered half-maps with a soft mask in Phenix[36], and then resampled to a pixel size of 0.756 Å.

For the dataset of TbAQP2-melarsoprol complex, 7,752,452 particles were extracted from 4,636 micrographs in cryoSPARC v.2 with a binning factor of 3 for ab initio reconstruction and heterogeneous refinement. The particles were subjected to one round of heterogeneous refinement in cryoSPARC v.2 to remove erroneously picked particles. Selected particles (1,907,100) were reextracted with a binning factor of 1 and subjected to several rounds of heterogeneous refinement (using C1 symmetry) to remove those particles that did not yield high-resolution reconstructions. After the heterogeneous refinement procedure, 580,515 particles were selected, reextracted, and subjected to 3D classification in RELION 3.1. 379,081 particles from the two best classes were selected and subjected to non-uniform refinement with C4 symmetry, which yielded a reconstruction at 2.8 Å overall resolution. After one round of Bayesian polishing in RELION 3.1, the particles were subjected to CTF (both global and local) and non-uniform refinements in cryoSPARC v.2, resulting in a map at 2.5 Å overall resolution. Another round of Bayesian polishing, and followed CTF (both global and local) and non-uniform refinement in cryoSPARC v.2 improved the resolution to 2.45 Å. The final reconstruction was further improved by density modification using the two unfiltered half-

maps with a soft mask in Phenix[36], and then resampled to the pixel size of 0.756 Å.

Atomic models were built de novo, refined in real space in COOT[37], and further refined in real space using PHENIX[38]. The final models have good stereochemistry and Fourier shell correlations (FSC) with the cryo-EM maps (Supplementary Fig. 2 to 4, and Table S1). Structural figures were prepared with Pymol (pymol.org), ChimeraX[39], Chimera[40], MOLE[41], CAVER[42], LIGPLOT[43], and HOLE[44].

## Molecular dynamic (MD) simulation
The OPM webserver was utilized to align the experimental structures within the lipid bilayer[45]. Membrane systems were constructed using the CHARMMGUI membrane building tool[46]. For protein and lipids, CHARMM36m force field parameters were used[47], while CGenFF parameters were used for the ligand[48]. The protein-ligand complexes were embedded in a lipid environment consisting of 128 POPC molecules, accompanied by 19507 TIP3P water molecules and 0.15 M NaCl, resulting in final systems containing 79485 atoms. MD simulations were conducted using GROMACS-2023.2[49]. The systems underwent an initial energy minimization step with 50000 steps employing the steepest descent algorithm. Subsequently, temperature equilibration was performed in the NVT ensemble at 310 K for 200 ps, followed by density equilibration in the NPT ensemble at 310 K and 1 atm for 10 ns. During the equilibration steps, heavy atoms were constrained using a harmonic restraint with a force constant of $1000\,kJ\,mol^{-1}\,nm^{-2}$. The cutoff for the nonbonded interactions was set to 10 Å, and long-range interactions were recovered by the particle mesh Ewald (PME) summation method[50]. The equilibrated system was then used for subsequent simulations.

## Cell-based TbAQP2 transport assays
The wild type and mutants of TbAQP2 were expressed in HEK293S GNTI- cells using the Bac-Mam system. In brief, recombinant baculovirus was generated in Spodoptera frugiperda Sf9 cells. After three rounds of amplification, P3 viruses were introduced to the culture when HEK293S GNTI- cells reached a density of $\sim\!3.0 \times 10^6$ cells/mL. Approximately 12–16 h post-transfection, 10 mM sodium butyrate was supplemented to the culture at 30 °C. Following an additional 60 h of incubation, cells were harvested by centrifugation and reconstituted to a concentration of $1 \times 10^6$ cells in Freestyle 293 media. Subsequently, cells were stained with 5 µmol/L stilbamidine (MedChemExpress, #HY-U00007A) at room temperature for 1 min. After washing and centrifugation, cells were resuspended in 1x PBS buffer and analyzed using a flow cytometer (Agilent Technologies, Inc., USA). All graphs were generated using FlowJo and GraphPad PRISM 6.0 software.

## Umbrella sampling (US) simulation
The cryo-EM structure of pentamidine-bound TbAQP2 was used as the starting structure. Steered molecular dynamics simulation (SMD) was carried out to generate the windows for the US simulation. The ligand was pulled along two directions separately, with one to the extracellular side and the other to the intracellular side. The distance between the current pentamidine position and that of pentamidine in the initial binding mode is used as the reaction coordinate (the distance for the initial state was set to 0.0 nm). For pulling pentamidine to the intracellular side, the distance is defined as positive, while that for pentamidine to the extracellular side is negative. A slow pulling rate (1 nm/100 ns) with a spring constant of $100\,kJ/mol/nm^2$ was used. Using the windows generated by SMD, a US simulation was performed, with the space between the two adjacent windows set to 0.1 nm. Two sets of US simulations were conducted, one featuring a static electrical field of 0.04 V/nm perpendicular to the membrane, and the other without such a field. Each window was simulated for 20 ns. The final analysis was performed with the weighted histogram analysis method

(WHAM), discarding the first 5 ns of each window[51]. After the analysis, the free-energy profiles of pentamidine to the inner and outer membrane were merged, using the same protocol as reported previously[52].

## MM/GBSA
The initial binding mode was determined via a clustering analysis protocol. Following this, the binding pose underwent a 5 ns-long unbiased MD simulation. To assess the effect of the W192A mutation on the binding mode's stability, we performed MM/GBSA calculations using the gmx_MMPBSA script[53]. The first 4 ns of the simulation trajectory were omitted from the analysis. Three independent simulations were performed.

## Reporting summary
Further information on research design is available in the Nature Portfolio Reporting Summary linked to this article.

## Data availability
The data that support this study are available from the corresponding authors upon request. Atomic coordinates and maps of the TbAQP2, TbAQP2-pentamidine complex, and TbAQP2-melarsoprol complex structures have been deposited in the PDB (accession numbers 8JY7, 8JY8, and 8JY6) and EMDB (EMD-36722, EMD-36723, and EMD-36721, respectively). Files relating to MD simulations (equilibrated system, parameter files for performing umbrella sampling, MD trajectories for the MM/GBSA analysis) have been uploaded to *Zenodo* [https://zenodo.org/records/11172197]. All data needed to evaluate the conclusions in the paper are present in the paper and/or the Supplementary Materials.

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

## Acknowledgements
We thank Stephen B. Long for the support on the initiation of this project; Haobo Pan, Ye Yu, Guoqiang Bi, Ning Jia, Houjun Xia, Wentao Hou, Huichao Ou, Ning Cui, Xiaoyu Liu, Qiuju Tang, Fanbin Zeng, Weidong Lin, Changlu Tao, and Xiaokang Zhang for the help with experiments; Single-particle cryo-EM data were collected at the Center of Cryo-Electron Microscopy at Southern University of Science and Technology. This work was supported by the National Natural Science Foundation of China (Grant No. 32370046 to C.W.; Grant No. 81671090 to Z.Z.) and China Postdoctoral Science Foundation (Grant No. 2023M733659 to W.C.).

## Author contributions
C.W. and Z.Z. conceived the project and designed the experiments. W.C. and Y.M. purified TbAQP2 and reconstituted it in lipid nanodisc. C.W. and W.C. generated cryo-EM samples and performed cryo-EM data collection, image processing, and model refinement. W.C., J.L., Y.X., and B.C. conducted cell-based transport assays. R.Z. performed molecular dynamic simulations with the assistance of J.Z. J.W. contributed to recombinant plasmid construction and protein purification. S.L. contributed to model refinement. C.W. wrote the manuscript with input from all the authors.

## Competing interests
The authors declare no competing interests.
