## [Peer Review File · Nature Communications]

Structural insights into drug transport by an aquaglyceroporinReviewer #1 (Remarks to the Author):

This study provides insights into the mechanism of pentamidine and melarsoprol drug entry into *Trypanosoma brucei*. Aquaporin 2 (AQP2) has been reported to be major facilitator of pentamidine and melarsoprol import into *T. Brucei*, an important step in their lethal disruption of protein translation. Various AQP2-dependent mechanisms for drug entry have been proposed. The first suggests that both drugs permeate the AQP2 channel by traversing its pore. This was considered unusual, since canonical aquaporin channel pores are exquisitely narrow and fine-tuned to permeate water. The second proposes a 'receptor'-mediated internalization where both drugs tightly bind to the extracellular side of AQP2 and the antimicrobial-bound AQP2 is endocytosed.

In this investigation, the authors delve deeper into these models by elucidating their cryo-EM structures in the presence or absence of each drug. Based on sequence homology, it was suggested that AQP2 likely possesses a wider ion permeation pathway due to the absence of canonical aquaporin 'ar/R motif', which sterically restricts the movement of larger solutes. Indeed, the TbAQP2 structure reveals a wider ($\geq \sim 4\text{\AA}$) pore diameter that is lined by smaller residues, which may accommodate larger organic antimicrobials such as pentamidine and melarsoprol.

Moreover, electron density indicates the presence of pentamidine and melarsoprol within the pore rather than in the proposed extracellular drug binding pocket. This strengthens the argument that TbAQP2 likely facilitates the internal permeation of these drugs into *Trypanosoma brucei* through a direct pore permeation mechanism. Subsequently, molecular dynamic simulations were conducted to further validate this proposed model. Overall, this is a very clear and insightful study with high-quality cryo-EM data collected and reasonable interpretation which contributes important information to the scientific community and adds weight to a mechanism of *T. brucei* antimicrobial drug entry.

Comments:

The only major comment that I have is the absence of experimental evidence to validate the authors' structural interpretation of the TbAQP2-pentamidine or TbAQP2-melarsoprol interactions that are critical. Much of the interpretation relied on prior publications and data, and although this interpretation is very reasonable, the novel ideas that came from the cryo-EM structures have yet to be experimentally explored.

Beyond this, I will also provide minor comments:

- Beyond its ability to conduct non-endogenous antimicrobials, why does AQP2 exist in *T. brucei*? Has its native function been elucidated? And if so, how do the structures contribute to our understanding of native TbAQP2 function?
- Within the first paragraph of the introduction, I would include one sentence that describes the overall the mechanism of action of these antimicrobials (preventing translation).
- This can be left to the authors discretion to include, but I think it would be worthwhile introducing the idea that *T. brucei* express 3 AQP paralogs, with AQP1 being the canonical water channel and AQP2 and 3 being non-canonical due to their lack of the ar/R motif, but that AQP2 appears to be solely responsible for pentamidine and melarsoprol uptake in *T. brucei* and this was evaluated by AQP2/3 putative pore swapping mutations.
- Line 52, replace the word "posit" to "propose" or something analogous
- Line 86-87. At the end of the sentence "However, TbAQP2 is an atypical aquaporin..." I would add a comment about how this is also unique to glycerol-permeable PfAQP. Maybe something like... "... canonical APQS and lacks the bulky residues observed in glycerol-permeable PfAQP."

- This is up to the discretion of the authors, but I would consider mentioning that D265, which was previously implicated in directly interacting with pentamidine in the "receptor-internalization model", is not within direct bonding distance in the cryo-EM structure and is instead forming a salt bridge with a neighboring arginine AQP2 residue (R269). This suggests that reduced pentamidine internalization in a TbAQP2 D265 mutant is likely due an allosteric effect that alters packing and electrostatic and hydrogen binding networks critical for maintaining the "fireman's grip" and promoting substrate binding and internalization.

- Line 131, the end of the sentence should read "a distinct permeation mechanism(s) for these antimicrobials."

- In Extended Data Fig 6, could the authors better highlight the hydrogen bond between N261 and S263.

- This is a merely an aesthetic suggestion, but in Figure 4D, I wonder if you can replace the arrows with something else (maybe red lightning bolts?) to indicate the clash.

- Line 182, replace the word "explains" with "supports"

- Line 200, "wild type of protein" should be "wild-type protein" or "wild-type TbAQP2"

Reviewer #2 (Remarks to the Author):

This manuscript reveals the structural insights of aquaporin (TbAQP2) on drug transport by means of molecular simulation and electron microscope experiment. The data are detailed and logical, and the "drug channel" hypothesis of TbAQP2 is confirmed to a certain extent, which has certain practical significance for guiding the design of new drugs for trypanosomiasis. Therefore, I think it is worth publishing, of course, I have a few questions.

1. Line 36 :

The author proposes that the cross-resistance of melarsol and pentamidine is due to the low absorption rate of the drug. So the author's logic is that he wants to further design new drugs by studying the mechanism of the transport of antitrypanosomic drugs by aquaporin TbAQP2, in order to improve the drug absorption rate? It is suggested that the author further explain the purpose and significance of introducing MPXR so that readers can understand it more clearly.

2. Line 174 :

The author mentions that the energy barrier of pentamidine to the cytoplasm is about 9 kcal/mol, which, combined with Figure 5, should be about 10 kcal/mol if the point where the curve intersects the horizontal axis is used as the energy barrier.

3. Line 200 :

The authors conclude that W192 is most likely to play a role as a docking site. Could one of the reasons for this conclusion be that W192 was observed interacting with extracellular vestibular amino formation in the captured structural snapshots? Is it considered to compare the binding energy of 1190 and W192 mutants with pentamidine?

4. Line 354 :

The authors mention that the temperature is balanced at 200ps in NVT and 10ns in NPT. However, as far as I know, the membrane system may also be unbalanced at tens of nanoseconds. Please explain the reasons why the author chose 200ps and 10ns for balance and prove that the system is balanced

at this time.

5.Line 372:

In the umbrella sampling simulation, one group is no electrostatic field, and the other group is 0.04V/nm electrostatic field perpendicular to the membrane. Please explain and supplement the reason why this electric field value is used.

RE: Structural insights into drug transport by an aquaglyceroporin

Authors: Wanbiao Chen, Rongfeng Zou, Yi Mei, Jiawei Li, Yumi Xuan, Bing Cui, Junjie Zou, Juncheng Wang, Shaoquan Lin, Zhe Zhang, Chongyuan Wang

General response to the reviewers: We thank the reviewers for their comments and feel that they have resulted in an improved manuscript. The comments have been addressed by additional experiments, revisions of the text, and by specific responses below (blue italics).

Reviewer #1 (Remarks to the Author):

This study provides insights into the mechanism of pentamidine and melarsoprol drug entry into *Trypanosoma brucei*. Aquaporin 2 (AQP2) has been reported to be major facilitator of pentamidine and melarsoprol import into *T. brucei*, an important step in their lethal disruption of protein translation. Various AQP2-dependent mechanisms for drug entry have been proposed. The first suggests that both drugs permeate the AQP2 channel by traversing its pore. This was considered unusual, since canonical aquaporin channel pores are exquisitely narrow and fine-tuned to permeate water. The second proposes a 'receptor'-mediated internalization where both drugs tightly bind to the extracellular side of AQP2 and the antimicrobial-bound AQP2 is endocytosed.

In this investigation, the authors delve deeper into these models by elucidating their cryo-EM structures in the presence or absence of each drug. Based on sequence homology, it was suggested that AQP2 likely possesses a wider ion permeation pathway due to the absence of canonical aquaporin 'ar/R motif', which sterically restricts the movement of larger solutes. Indeed, the TbAQP2 structure reveals a wider ($\geq \sim 4\text{\AA}$) pore diameter that is lined by smaller residues, which may accommodate larger organic antimicrobials such as pentamidine and melarsoprol.

Moreover, electron density indicates the presence of pentamidine and melarsoprol within the pore rather than in the proposed extracellular drug binding pocket. This strengthens the argument that TbAQP2 likely facilitates the internal permeation of these drugs into *Trypanosoma brucei* through a direct pore permeation mechanism. Subsequently, molecular dynamic simulations were conducted to further validate this proposed model. Overall, this is a very clear and insightful study with high-quality cryo-EM data collected and reasonable interpretation which contributes important information to the scientific community and adds weight to a mechanism of *T. brucei* antimicrobial drug entry.

We thank the reviewer for appreciating the quality and novelty of our work.

Comments:

The only major comment that I have is the absence of experimental evidence to validate the authors' structural interpretation of the TbAQP2-pentamidine or TbAQP2-melarsoprol

interactions that are critical. Much of the interpretation relied on prior publications and data, and although this interpretation is very reasonable, the novel ideas that came from the cryo-EM structures have yet to be experimentally explored.

We appreciate this important comment. We have revised the text and conducted cell-based uptake assays to address it.

The function of TbAQP2 was reconstituted in HEK293 cells using the fluorescent analogue of pentamidine (DASPI) as a substrate. In contrast to uninfected HEK293 cells, those infected with TbAQP2 displayed robust uptake of DASPI (approximately 60% positive) within 1 minutes (see Extended Data Fig. 1a, b). Mutagenesis of the substrate-binding residues led to significant decreases (3 to 80-fold) in TbAQP2-dependent uptake of DASPI (see Extended Data Fig. 9). Overall, the functional data suggest that the key residues involved in pentamidine and melarsoprol interaction are crucial for the function of TbAQP2, which aligns with our structural observations.

Beyond this, I will also provide minor comments:

- Beyond its ability to conduct non-endogenous antimicrobials, why does AQP2 exist in *T. brucei*? Has its native function been elucidated? And if so, how do the structures contribute to our understanding of native TbAQP2 function?

We agree that the native functions of TbAQP2 are very interesting. It has been reported that TbAQP1, TbAQP2, and TbAQP3 function in the excretion of unwanted metabolic end-products, including methylglyoxal, D-lactate, L-lactate, and acetate (Uzcátegui et al., Biochim Biophys Acta Biomembr 2018, PMID: 30409521). However, the exclusive physiological function of TbAQP2 remains unclear.

- Within the first paragraph of the introduction, I would include one sentence that describes the overall the mechanism of action of these antimicrobials (preventing translation).

Thank you for this suggestion. We have added these sentences in the introduction of the revised manuscript.

- This can be left to the authors discretion to include, but I think it would be worthwhile introducing the idea that *T. brucei* express 3 AQP paralogs, with AQP1 being the canonical water channel and AQP2 and 3 being non-canonical due to their lack of the ar/R motif, but that AQP2 appears to be solely responsible for pentamidine and melarsoprol uptake in *T. brucei* and this was evaluated by AQP2/3 putative pore swapping mutations.

Thank you for this suggestion. We have added these sentences in the introduction of the revised manuscript.

- Line 52, replace the word “posit” to “propose” or something analogous

Thank you. We have made the modification.

- Line 86-87. At the end of the sentence “However, TbAQP2 is an atypical aquaporin...” I would add a comment about how this is also unique to glycerol-permeable PfAQP. Maybe something like... “... canonical APQS and lacks the bulky residues observed in glycerol-permeable PfAQP.”

Thank you for this suggestion. We have added this sentence in the revised manuscript.

- This is up to the discretion of the authors, but I would consider mentioning that D265, which was previously implicated in directly interacting with pentamidine in the “receptor-internalization model”, is not within direct bonding distance in the cryo-EM structure and is instead forming a salt bridge with a neighboring arginine AQP2 residue (R269). This suggests that reduced pentamidine internalization in a TbAQP2 D265 mutant is likely due an allosteric effect that alters packing and electrostatic and hydrogen binding networks critical for maintaining the “fireman’s grip” and promoting substrate binding and internalization.

Thank you for this suggestion. We have added sentences and a figure (Extended Data Fig. 10) to mention the D265 residue observed in the cryo-EM structures in the revised manuscript.

- Line 131, the end of the sentence should read “a distinct permeation mechanism(s) for these antimicrobials.”

Thank you. We have corrected this.

- In Extended Data Fig 6, could the authors better highlight the hydrogen bond between N261 and S263.

Thank you for this suggestion. We have made the modification in the revised Extended Data Fig 6.

- This is a merely an aesthetic suggestion, but in Figure 4D, I wonder if you can replace the arrows with something else (maybe red lightning bolts?) to indicate the clash.

Thank you for this suggestion. We have made the modification in the revised Figure 4D.

- Line 182, replace the word “explains” with “supports”

Thank you. We have corrected this.

- Line 200, “wild type of protein” should be “wild-type protein” or “wild-type TbAQP2”

Thank you. “wild type of protein” has been replaced with “wild-type protein”.

Reviewer #2 (Remarks to the Author):

This manuscript reveals the structural insights of aquaporin (TbAQP2) on drug transport by means of molecular simulation and electron microscope experiment. The data are detailed and logical, and the "drug channel" hypothesis of TbAQP2 is confirmed to a certain extent, which has certain practical significance for guiding the design of new drugs for trypanosomiasis. Therefore, I think it is worth publishing, of course, I have a few questions.

We thank the reviewer for appreciating the quality and novelty of our work.

1. Line 36:

The author proposes that the cross-resistance of melarsol and pentamidine is due to the low absorption rate of the drug. So the author's logic is that he wants to further design new drugs by studying the mechanism of the transport of antitrypanosomic drugs by aquaporin TbAQP2, in order to improve the drug absorption rate? It is suggested that the author further explain the purpose and significance of introducing MPXR so that readers can understand it more clearly.

Thank you for this suggestion. We hope our work can help guide the design of new drugs against trypanosomiasis. The significance of MPXR has been highlighted in the revised manuscript.

2. Line 174:

The author mentions that the energy barrier of pentamidine to the cytoplasm is about 9 kcal/mol, which, combined with Figure 5, should be about 10 kcal/mol if the point where the curve intersects the horizontal axis is used as the energy barrier.

Response: Thank you for pointing out the error. We changed to 10 kcal/mol in the revised manuscript.

3. Line 200:

The authors conclude that W192 is most likely to play a role as a docking site. Could one of the reasons for this conclusion be that W192 was observed interacting with extracellular vestibular amino formation in the captured structural snapshots? Is it considered to compare the binding energy of I190 and W192 mutants with pentamidine?

Thank you for this suggestion. The meta-stable state identified during the binding process indicates that W192 engages in pi-pi interactions with pentamidine. This metastable state is less stable than the wild type (WT) in the W192A variant, as demonstrated in the MM/GBSA simulation. As for residue I190, it does not directly interact with pentamidine. However, I190's micro-environment is hydrophobic, and its mutation to Thr is likely to alter the protein structure. This alteration could potentially lead to a reduced binding affinity for pentamidine.

Figure 1: Residue W192 involving pentamidine binding and permeation. Cartoon representation of the local minima on the protein-ligand binding path. Pentamidine (green), W192 (magenta) I190 (cyan) were shown in sticks.

4.Line 354:

The authors mention that the temperature is balanced at 200ps in NVT and 10ns in NPT. However, as far as I know, the membrane system may also be unbalanced at tens of nanoseconds. Please explain the reasons why the author chose 200ps and 10ns for balance and prove that the system is balanced at this time.

Thank you for this question. In the NVT equilibration step, it is essential to monitor the system's temperature as a function of simulation time. Figure 1a illustrates that the temperature stabilized swiftly at 310 K, signifying that the system attained the target temperature efficiently. During the NPT ensemble equilibration, the pressure is the parameter of interest. As depicted in Figure 1b, although the pressure exhibits significant fluctuations over time, which is common in the NPT equilibration step, it fluctuates around the desired value (1 bar).

For membrane proteins, membrane thickness and area per lipid are critical indicators of system equilibrium. We compared these parameters with a 100-ns production MD simulation. Figures 2a and 2b show that the membrane thickness during the last 2 ns of the NPT equilibration averaged 3.82 ± 0.08 nm, closely aligning with the 3.76 ± 0.09 nm observed in the production MD simulation. Similarly, Figures 2c and 2d reveal that the area per lipid for the last 2 ns of the NPT equilibration was 0.69 ± 0.03 nm², comparable to the 0.70 ± 0.03 nm² obtained in the production MD.

Considering the consistency of membrane thickness and area per lipid between the NPT equilibration and production MD, we conclude that the system has achieved equilibrium in the NPT ensemble.

Figure 2: a) Changes in temperature with respect to simulation time in the NVT equilibration and b) time evolution of pressure of the simulated system.

Figure 3: Changes in membrane thickness with respect to simulation time in a) NPT equilibration and b) 100-ns MD. Temporal evolution of area per lipid in c) NPT equilibration and d) 100-ns MD.

5.Line 372:

In the umbrella sampling simulation, one group is no electrostatic field, and the other group is 0.04V/nm electrostatic field perpendicular to the membrane. Please explain and supplement the reason why this electric field value is used.

Thank you for this question. The membrane potential ($\Delta\psi$) is calculated by multiplying the electrostatic field (E) by the thickness of the membrane (d), as expressed by the equation $\Delta\psi = -E \times d$. With a membrane thickness of 3.76 nm and an electrostatic field of 0.04 V/m, the

resulting $\Delta\psi$ is -150.4 mV. This value is in proximity to the experimentally determined membrane potential for pentamidine uptake in trypanosomes, which is -125 mV for *T. brucei*.

Reviewer #1 (Remarks to the Author):

Many thanks to the authors for addressing all of my suggestions. I do not have any further comments.

Reviewer #2 (Remarks to the Author):

I am fully satisfied with the answers to my questions as well as with the modifications of the manuscript. I have no further comments and thus I recommend the manuscript for the publication.

RE: Structural insights into drug transport by an aquaglyceroporin

Authors: Wanbiao Chen, Rongfeng Zou, Yi Mei, Jiawei Li, Yumi Xuan, Bing Cui, Junjie Zou, Juncheng Wang, Shaoquan Lin, Zhe Zhang, Chongyuan Wang

2nd round of revision

Reviewer #1 (Remarks to the Author): Many thanks to the authors for addressing all of my suggestions. I do not have any further comments.

We thank the reviewers for their comments and feel that they have resulted in an improved manuscript.

Reviewer #2 (Remarks to the Author): I am fully satisfied with the answers to my questions as well as with the modifications of the manuscript. I have no further comments and thus I recommend the manuscript for the publication.

We thank the reviewers for their comments and feel that they have resulted in an improved manuscript.